# A Retrospective Study on the Use of Daptomycin and Linezolid in Singapore General Hospital

**DOI:** 10.3390/antibiotics14111088

**Published:** 2025-10-28

**Authors:** Boon San Teoh, Yi Xin Liew, Yibo Wang, Shimin Jasmine Chung, Ban Hock Tan

**Affiliations:** 1Department of Pharmacy, Singapore General Hospital, Outram Road, Singapore 169608, Singapore; teoh.boon.san@sgh.com.sg (B.S.T.); liew.yi.xin@sgh.com.sg (Y.X.L.); wang.yibo@sgh.com.sg (Y.W.); 2Singhealth Duke-NUS Medicine Academic Clinical Programme, 8 College Road, Level 4, Singapore 169857, Singapore; jasmine.chung.s.m@singhealth.com.sg; 3Department of Infectious Diseases, Singapore General Hospital, Outram Road, Singapore 169608, Singapore; 4Singhealth Duke-NUS Transplant Center, Singapore 169608, Singapore

**Keywords:** vancomycin-resistance, antimicrobial stewardship, daptomycin, linezolid

## Abstract

**Background:** Vancomycin-resistant Enterococcus (VRE) has emerged as a major nosocomial pathogen. A recent surveillance of our hospital identified a concerning rise in VRE bacteremia since 2020, despite the stable use of broad-spectrum antibiotics. This trend, coupled with the increased use of daptomycin and linezolid for drug-resistant Gram-positive bacteremia (GPB), prompted an evaluation of their usage beyond approved hospital indications. **Methods:** A retrospective analysis was carried out from 1 February 2023 to 31 July 2023, during which 100 and 195 patients received linezolid and daptomycin, respectively. Patients’ data were extracted from the hospital’s electronic medical records, and the appropriateness of the antibiotics prescribed was assessed. The amount of daptomycin and linezolid utilization during the study period was also retrieved, as was the incidence of VRE bacteremia. **Results:** A total of 295 courses of VRE-active agents, linezolid (*n* = 100) and daptomycin (*n* = 195), were assessed for appropriateness in this study. Linezolid and daptomycin use were judged as inappropriate 5.0% and 9.2% of the time, respectively. The primary reason for inappropriate linezolid use was overly broad empirical therapy where first-line options like cefazolin and vancomycin could have been prescribed. Daptomycin was often used inappropriately in non-VRE infections, and surgical prophylaxis or use was extended unnecessarily without microbiological justification. **Conclusions:** Linezolid and daptomycin were prescribed appropriately. Nevertheless, our findings suggest the need to re-evaluate the empirical treatment strategies especially in VRE-colonized patients. Implementation of robust risk-based criteria as well as in-house hospital guidelines or protocols on the initiation of VRE-active agents may help support more judicious prescribing practices of these agents.

## 1. Introduction

Vancomycin-resistant Enterococcus (VRE) has emerged as a significant nosocomial pathogen worldwide, with prevalence rates varying from 5 to 30% in different healthcare settings [1,2]. VRE infections are associated with increased mortality, prolonged hospital stays, and substantial healthcare costs compared to vancomycin-susceptible enterococcal infections [3,4]. The emergence of VRE has necessitated the use of alternative antimicrobial agents, primarily daptomycin and linezolid, which represent the cornerstone of VRE treatment [5,6].

Daptomycin, a cyclic lipopeptide antibiotic, demonstrates rapid bactericidal activity against VRE with minimal resistance development [7]. Linezolid, an oxazolidinone antibiotic, offers the advantages of excellent oral bioavailability and tissue penetration, making it suitable for both intravenous and oral therapies [8]. However, both agents are associated with significant costs and potential adverse effects, including dose-dependent peripheral neuropathy and thrombocytopenia with linezolid and creatine phosphokinase elevation with daptomycin [9,10].

The appropriate use of these VRE-active agents is crucial for antimicrobial stewardship programs. Studies have reported inappropriate prescribing rates ranging from 10 to 25% for these agents, often due to unnecessary empirical use, inappropriate duration, or failure to de-escalate therapy when cultures become available [11,12]. Risk factors for VRE colonization and subsequent infection include prolonged hospitalization, intensive care unit admission, immunocompromised status, and prior antibiotic exposure, particularly to vancomycin [13,14].

Antimicrobial stewardship programs have traditionally focused on broad-spectrum beta-lactam antibiotics to combat resistance in Gram-negative organisms such as extended-spectrum β-lactamases (ESBL) producers [15,16]. However, the increasing prevalence of drug-resistant Gram-positive infections, particularly VRE, necessitates expanded stewardship efforts to include monitoring of VRE-active agents [17].

While antimicrobial stewardship studies on VRE-active agents exist globally, few have examined the prescribing patterns in settings with pre-existing specialist restrictions. Singapore’s General Hospital mandates infectious diseases physician approval before prescribing both daptomycin and linezolid, representing a more restrictive environment than that in many reported studies [18,19]. This restriction model aligns with the antimicrobial stewardship principles advocated across Asia–Pacific healthcare systems, where rising antimicrobial resistance has prompted the implementation of stringent prescribing control [20,21]. Additionally, the concurrent rises in VRE incidence and VRE-active agent use in our institution present a unique context for evaluating prescribing appropriateness, as it raises questions about whether increased use represents an appropriate clinical response or potential overuse through inappropriate empirical prescribing—a phenomenon less-studied in highly regulated prescribing environments.

In our hospital, surveillance data revealed a concerning increase in VRE bacteremia incidence since 2020, with rates per 10,000 patient days demonstrating an upward trend from January 2019 to September 2022, peaking in August 2020 [data not published]. This increase occurred despite stable use of broad-spectrum antibiotics and was accompanied by a parallel rise in daptomycin and linezolid use. The temporal relationship between rising VRE incidence and increased consumption of VRE-active agents raised important questions about prescribing appropriateness, as increased use could represent either an appropriate clinical response to rising VRE infections or potential overuse through inappropriate empirical prescribing.

Therefore, we conducted this retrospective study to evaluate the appropriateness of daptomycin and linezolid prescribing practices in our institution and identify opportunities for antimicrobial stewardship improvement.

## 2. Results

### 2.1. Patient Demographics and Baseline Characteristics

A total of 295 antibiotic courses were analyzed: 100 linezolid courses and 195 daptomycin courses prescribed between February and July 2023. Patient demographics and comorbidities are presented in Table 1. The median age was similar between groups (68 years for linezolid vs. 69 years for daptomycin). Patients receiving daptomycin had higher rates of chronic kidney disease (43.6% vs. 29%) and hematological malignancies (11.3% vs. 3%), whilst those receiving linezolid were more likely to be solid-organ transplant recipients (4% vs. 0%).

### 2.2. Linezolid Prescribing Patterns and Clinical Indications

Linezolid was predominantly used as oral therapy, with 75% of courses administered via the oral route, reflecting its role as a step-down agent. The majority of linezolid prescriptions were for culture-directed therapy (73%, *n* = 73), followed by empirical therapy (25%, *n* = 25) and prophylaxis (2%, *n* = 2).

Among patients receiving empirical linezolid, 40% (10/25) were VRE-colonized. The culture yield for VRE among these patients was low; none of the patients colonized with VRE who were treated empirically with linezolid subsequently grew VRE (0%, 95% CI: 0.0–27.8% [0/10]). Among patients who were non-VRE-colonized and receiving empirical linezolid, VRE was isolated in only one case (6.7%, 95% CI: 1.2–29.8% [1/15]). This difference was not statistically significant (*p* = 1.0).

### 2.3. Daptomycin Prescribing Patterns and Clinical Indications

Daptomycin use was more evenly distributed between culture-directed therapy and empirical therapy. Of the 195 courses, 48.2% (*n* = 94) were culture-directed, and 47.7% (*n* = 93) were empirical, with a small proportion used for prophylaxis (4.1%, *n* = 8).

Among patients receiving empirical daptomycin, VRE colonization was common, occurring in 69.9% (65/93) of cases. Within this colonized subgroup, 64.6% (*n* = 42) were immunocompromised, and 43.1% (*n* = 28) presented with hemodynamic instability requiring vasopressor support or intensive care admission. Despite these risk factors, the culture yield for VRE remained low. Among the patients colonized with VRE who were treated empirically with daptomycin, VRE was subsequently isolated in 7.7%, 95% CI: 3.3–16.8% [5/65]. Among patients not colonized with VRE and receiving empirical daptomycin, VRE was not isolated in any case (0%, 95% CI: 0.0–12.1% [0/28]). This difference was not statistically significant (*p* = 0.32).

### 2.4. Appropriateness Assessment Results

Overall appropriateness rate: The majority of prescriptions were deemed appropriate. Inappropriate use occurred in 5.0% [95% CI: 0.7–9.3%] (5/100) of linezolid courses and 9.2% [95% CI: 5.1–13.3%] (18/195) of daptomycin courses. This difference was not statistically significant (*p* = 0.25).

Reasons for inappropriate linezolid use *(n* = 5): All inappropriate linezolid courses were due to inappropriate antibiotic selection, where narrower-spectrum alternatives were available: perioperative prophylaxis for pacemaker insertion where cefazolin was indicated (*n* = 1), empirical treatment of skin and soft tissue infections where cefazolin could have been used (*n* = 2), response to preliminary blood culture results showing Gram-positive cocci in clusters in a patient colonized with VRE before organism identification and susceptibility testing (*n* = 1), and empirical use for intra-abdominal abscess in a clinically stable patient (*n* = 1).

Reasons for inappropriate daptomycin use (*n* = 18): Inappropriate daptomycin use was due to both inappropriate selection (*n* = 16) and inappropriate duration (*n* = 2). Selection issues included surgical prophylaxis where standard agents (e.g., vancomycin) were appropriate (*n* = 3), empirical use in patients not colonized with VRE with various infections where vancomycin could have been used (*n* = 11), and empirical use in patients who were VRE-colonized but hemodynamically stable (*n* = 2). Duration issues involved continuation of daptomycin despite culture results showing *Klebsiella pneumoniae* monomicrobial bacteremia in one patient and vancomycin-susceptible *Enterococcus faecium* bacteremia in another (*n* = 2).

### 2.5. Factors Associated with Inappropriate Use

The sources of infection and a detailed breakdown of inappropriate courses are presented in Table 2. Common reasons for empirical use in patients who were not colonized with VRE are shown in Table 3, with avoidance of the adverse effects associated with vancomycin being the predominant reason for daptomycin selection (58.8%) and intravenous-to-oral conversion being the main driver for linezolid use (50%). Additional details on the rationale for prescribing linezolid or daptomycin in patients without vancomycin-resistant Enterococcus (VRE) or other vancomycin-intermediate/resistant Gram-positive organisms, as well as the pathogens isolated from clinical cultures for which these agents were used, are provided in the Appendix A.

## 3. Discussion

This audit evaluating the appropriate selection of linezolid and daptomycin revealed several important findings regarding the utilization patterns at our institution. While overall inappropriate use rates were relatively low (5.0% for linezolid and 9.2% for daptomycin; *p* = 0.25), our findings highlight potential areas for antimicrobial stewardship improvement and provide insights that both align with and contrast those in the existing literature.

### 3.1. Linezolid

Linezolid was predominantly prescribed as culture-directed therapy, accounting for 73% of courses, while only 25% were empirical. Among patients receiving empirical linezolid, 40% (10/25) were VRE-colonized; however, none were subsequently colonized with VRE. In patients who were non-VRE-colonized, the yield was similarly low at 6.7% (1/15). These findings suggest that empirical VRE coverage with linezolid may be unnecessary in most cases, which aligns with recent evidence questioning the routine use of empirical VRE therapy [14,22]. Our low culture yield supports these studies’ conclusions that empirical VRE treatment may expose patients to unnecessary drug-related toxicities without clinical benefit.

A notable stewardship success was the high rate of intravenous-to-oral conversion, with 75% of linezolid courses administered orally. This aligns with emerging evidence supporting oral linezolid for serious infections. The POET trial established oral linezolid as effective for left-sided infective endocarditis in stable patients [23], while Dagher et al. demonstrated its utility as step-down therapy for uncomplicated Staphylococcus aureus bacteremia [24]. Our practice appears consistent with these evidence-based approaches. Inappropriate linezolid use (*n* = 5) primarily involved antibiotic selection where narrower-spectrum alternatives were available, including perioperative prophylaxis for pacemaker insertion, empirical treatment of skin and soft tissue infections, and premature response to preliminary culture results.

### 3.2. Daptomycin

Daptomycin was prescribed nearly twice as often as linezolid (195 versus 100 courses) and was evenly distributed between empirical treatment (47.7%) and culture-directed therapy (48.2%). Among patients receiving empirical daptomycin, 69.9% (65 of 93) were VRE-colonized, with 42 (64.6%) being immunocompromised and 28 (43.1%) presenting with hemodynamic instability. However, VRE was subsequently isolated in only 7.7% (5/65) of patients who were VRE-colonized and treated with empirical daptomycin. This 7.7% incidence is substantially lower than previously reported rates of 19–33% for patients with clinically significant VRE infections [14,22]. Furthermore, Kamboj et al. demonstrated that empirical VRE therapy in patients who were colonized and immunocompromised did not improve treatment duration or mortality outcomes [25], while Snyder et al. similarly found no mortality benefit from empirical VRE coverage in hematology patients [26]. Similar to linezolid, the yield among patients who were non-VRE-colonized was extremely low: 0% (0/28) for daptomycin, indicating that empirical VRE coverage in this population may be unnecessary.

Inappropriate daptomycin use (*n* = 18) was due to both inappropriate selection (*n* = 16) and inappropriate duration (*n* = 2). Selection issues predominantly involved empirical use in patients who were non-VRE-colonized (*n* = 11) where vancomycin could have been used, surgical prophylaxis (*n* = 3), and empirical use in patients who were VRE-colonized but hemodynamically stable (*n* = 2). Duration issues involved continuation despite culture results showing *Klebsiella pneumoniae* monomicrobial bacteremia and vancomycin-susceptible *Enterococcus faecium* bacteremia. Avoidance of vancomycin-associated adverse effects was the predominant reason for daptomycin selection (58.8%), reflecting legitimate clinical considerations, as vancomycin-associated nephrotoxicity remains clinically relevant despite improved formulations, particularly with concomitant nephrotoxic agents [27,28,29], which are common in our patient population.

### 3.3. General Overview and Limitations

The prescribing patterns between daptomycin and linezolid showed notable differences. Daptomycin was used nearly twice as frequently as linezolid (195 vs. 100 courses), and its use was evenly distributed between empirical treatment and culture-directed therapy. In contrast, linezolid was predominantly used as culture-directed therapy (73% of courses), with only 25% being empirical. A significant proportion of our patients had hematological malignancies and were neutropenic, which may explain the preference for daptomycin over linezolid due to the risk of myelosuppression associated with the latter.

Our study provides unique insights into VRE-active agent prescribing in a highly restricted environment. Despite mandatory infectious disease physician oversight—a more stringent control—we still observed 5–9% inappropriate use. This finding suggests that even specialist-level restrictions may not eliminate inappropriate prescribing entirely, highlighting the need for additional stewardship interventions beyond prescriber restrictions alone. This contrasts those of studies from less-restrictive environments where inappropriate use rates of 15–30% have been reported for reserve antibiotics [30,31], suggesting our restriction model provides substantial benefit while identifying areas for further improvement.

Although clinical outcomes were not directly assessed, prescribing patterns indicate generally safe and targeted use of VRE-active agents. The low incidence of inappropriate prescribing and increased adoption of oral linezolid step-down therapy likely contributed to improved patient outcomes and cost savings. These findings highlight key opportunities to strengthen antimicrobial stewardship, including establishing stricter criteria for initiating empirical VRE coverage in patients who are colonized, implementing robust de-escalation and discontinuation protocols upon receipt of culture results, and developing clear guidance for surgical prophylaxis in individuals who are VRE-colonized. Furthermore, enhanced monitoring of therapy duration, supported by prospective audit and feedback, could further optimize antimicrobial utilization.

This study has several limitations. Its retrospective design, which relied on electronic medical record documentation, may have led to incomplete capture of prescriber rationale. The single-center setting and relatively short six-month duration limit external validity and may not fully represent long-term prescribing patterns. Additionally, the study period did not account for seasonal variations in infection rates or outbreak dynamics, which could influence prescribing behavior and antimicrobial utilization. Outpatient oral linezolid prescriptions were excluded, although this likely reflected continuation of inpatient therapy. Finally, clinical outcomes such as mortality and length of stay were not assessed, and no cost analysis of inappropriate antimicrobial use was performed.

## 4. Methods

A retrospective analysis was carried out on all patients who were prescribed daptomycin and linezolid (both oral and intravenous) during their admission from February 2023 to July 2023. A patient list was extracted from the hospital’s data warehouse. Data collected included age, gender, admission and discharge dates, audited antibiotic start and stop dates, as well as duration of treatment. Appropriateness of the use of these antibiotics was evaluated through the patients’ electronic medical records, based on indication for use, source of infection, and duration. Appropriateness assessments were conducted by a multidisciplinary team consisting of two infectious disease physicians and two clinical pharmacists with expertise in antimicrobial stewardship. All assessors independently reviewed each case, and disagreements were resolved through consensus discussion.

A course of daptomycin or linezolid was defined as the period of continued use of the antibiotic. For example, daptomycin that was stopped ≥48 h but subsequently restarted was considered two courses. Standard therapy consisted of oral or intravenous linezolid at 600 mg every 12 h and daptomycin at 6–10 mg/kg every 24 h, with appropriate dose adjustments based on renal function. Dosing appropriateness was not assessed in this audit as both daptomycin and linezolid dosing protocols are standardized in our institution according to renal function and indication, with mandatory infectious disease physician oversight ensuring appropriate dosing decisions. Our focus was primarily on indication appropriateness and duration, which represent the main areas of concern identified in our preliminary stewardship reviews.

Statistical analysis was performed using descriptive statistics for baseline characteristics and clinical outcomes. Categorical variables were compared using Fisher’s exact test. All tests were two-sided, and a *p*-value of <0.05 was considered statistically significant. These exploratory statistical analyses were conducted using [specify software, e.g., SPSS version 26, IBM Corp., Armonk, NY, USA].

This study was approved by the hospital’s Institutional Review Board (CIRB Ref: 2022/2560). For the purpose of this audit, we classified the primary indication for daptomycin, and linezolid use based on the documented clinical rationale and microbiological context. Anti-VRE indication was assigned when (1) VRE was isolated from clinical cultures, (2) empirical VRE coverage was explicitly documented in patients who were VRE-colonized, or (3) broad vancomycin-resistant Gram-positive coverage was specifically mentioned in clinical notes when treating suspected resistant enterococcal or staphylococcal infections. Anti-Methicillin-resistant Staphylococcus aureus (MRSA) indication was assigned for documented MRSA infections or when MRSA coverage was the primary stated indication. Cases where both indications were present were classified based on the predominant clinical concern as documented by the treating physician. This classification was validated through a review of clinical notes, culture results, and consultation recommendations. The use of linezolid and daptomycin was considered appropriate if indication for use was one of the following:Culture-directed therapy.An alternative to vancomycin due to patient allergy status or to avoid vancomycin-associated adverse drug events such as nephrotoxicity.Perceived or true difficulty in achieving therapeutic vancomycin trough levels.For treatment of complicated MRSA infections not responding to vancomycin (e.g., lack of clinical improvement after 48–72 h of appropriate vancomycin therapy, persistent bacteremia despite adequate vancomycin levels, or deep-seated infections with poor vancomycin penetration).An oral alternative, in the case of linezolid, for treatment of Gram-positive infections in patients with poor venous access.For outpatient parenteral antibiotic therapy (OPAT).For empirical VRE coverage in patients who were VRE-colonized who were neutropenic or immunocompromised and experiencing hemodynamic instability.For empirical coverage in patients colonized with VRE or other vancomycin-resistant Gram-positive bacteria who ha clinically deteriorated despite broad spectrum antibiotics.

Appropriate duration was defined based on established clinical guidelines and institutional protocols: 7–14 days for uncomplicated bacteremia [32,33], 4–6 weeks for endocarditis [34], 6–8 weeks for osteomyelitis [35], and individualized duration based on clinical response and source control for other infections. Duration was considered inappropriate when therapy was continued beyond recommended timeframes without clear clinical justification or microbiological evidence.

Both daptomycin and linezolid are included in our hospital formulary as restricted antibiotics requiring infectious disease physician approval for initiation. These agents can only be prescribed by infectious disease specialists or following consultation with infectious disease physicians, with emergency use permitted in after-hours scenarios requiring retrospective infectious disease physician review within 24 h. This restriction policy was implemented to ensure appropriate utilization of these high-cost anti-VRE agents and to minimize resistance development.

### Institutional Context and Infection Control Measures

During the study period, no formal VRE or MRSA outbreaks were declared according to institutional outbreak definitions. However, our hospital maintains active surveillance for VRE colonization in high-risk patients, including those admitted to intensive care units; high-dependency, intermediate-care areas; renal, hematology–oncology wards; patients with prior VRE history, history of hospitalization in overseas or local private or local public hospital in the past 1 year, and long stays (>14 days). VRE screening was performed using rectal swabs processed through selective chromogenic agar media, with molecular confirmation for vancomycin-resistance genes (vanA/vanB) when required. Patients colonized with VRE were managed under contact precautions, with dedicated nursing care and enhanced environmental cleaning protocols.

## 5. Conclusions

This audit identified that the uses of daptomycin and linezolid in our hospital were mostly appropriate. However, we recommend re-evaluating empirical treatment strategies for patients who are VRE-colonized and continue to prioritize individualized patient factors despite the push toward the use of narrow-spectrum antibiotics to balance efficacy and safety. Also, future studies should focus on developing and validating risk stratification tools for VRE infection in patients who are colonized and evaluating the impact of more-stringent empiric therapy guidelines on patient outcomes and antimicrobial resistance patterns. Prospective studies comparing different approaches to empirical therapy in patients who are VRE-colonized would be particularly valuable in guiding future practice.

## Figures and Tables

**Table 1 antibiotics-14-01088-t001:** Demographic data.

Variable	Linezolid (*n* = 100)	Daptomycin (*n* = 195)
Age, years, median (IQR)	68 (60–75)	69 (59–76)
Sex		
Female	39 (39%)	89 (45.6%)
Male	61 (61%)	106 (54.4%)
**Comorbidities**		
Patients with ≥1 comorbidity *	71 (71%)	173 (88.7%)
Diabetes	40 (40%)	79 (40.5%)
Chronic kidney disease	29 (29%)	85 (43.6%)
Bone marrow transplant recipient	0 (0%)	18 (9.2%)
Solid-organ transplant recipient	4 (4%)	0 (0%)
Solid tumor	30 (30%)	59 (30.3%)
Hematological malignancy	3 (3%)	22 (11.3%)

*n* = number of cases; * one patient could have had more than one comorbidity.

**Table 2 antibiotics-14-01088-t002:** Summary of indication, number of inappropriateness courses, and source of infection.

Anti-VRE Agent	Linezolid (*n* = 100)			Daptomycin (*n* = 195)		
**Indication**	Prophylaxis (*n* = 2)	Empiric (*n* = 25)	Culture-Directed (*n* = 73)	Prophylaxis (*n* = 8)	Empiric (*n* = 93)	Culture-Directed (*n* = 94)
Number of inappropriate courses of antibiotics ^a^	1 (choice)	4 (choice)	NA	3 (choice)	13 (choice), 2 (duration)	NA
*** Source of infection**						
Bone and joint	N.A.	3	1	N.A.	4	9
Cardiovascular system	N.A.	0	0	N.A.	2	0
Central nervous system	N.A.	1	0	N.A.	0	0
EENT	N.A.	1	1	N.A.	2	3
Gastrointestinal system	N.A.	3	11	N.A.	32	27
Genitourinary		1	10	N.A.	6	17
Lower respiratory tract/pneumonia	N.A.	7	7	N.A.	0	0
Skin and soft tissue	N.A.	16	31	N.A.	15	31
Febrile neutropenia	N.A.	0	0	N.A.	7	0
CRBSI	N.A.	3	13	N.A.	8	12
Unknown source	N.A.	1	0	N.A.	21	0

EENT: eye, ear, nose, throat, or mouth; CRBSI: catheter-related bloodstream infection; N.A.: not applicable; *n* = number of cases; ^a^ inappropriate course of antibiotics is based on preset criteria defined in the text above. * One patient may have had ≥1 site of infection.

**Table 3 antibiotics-14-01088-t003:** Common reasons for appropriate empirical use of linezolid and daptomycin in patients who were not colonized with VRE.

No	Reasons	Number of Cases/(%) ^#^	Examples
Linezolid			
1.	IV-to-PO switch	7 (50)	▪ High bioavailability of PO linezolid
2.	Avoid therapeutic drug monitoring (TDM)	5 (35.7)	▪ Avoid frequent TDM with vancomycin
3.	Avoid adverse drug reaction (ADR) from first-line antibiotics	2 (14.3)	▪ Avoidance of nephrotoxicity
	**Total number**	14	
**Daptomycin**			
1.	Avoid frequent TDM and for use in outpatient parental antibiotic therapy (OPAT)	2 (11.8)	▪ Avoid frequent trough monitoring with vancomycin
2.	Avoid ADR from first-line antibiotics	10 (58.8)	▪ Avoid nephrotic agents in pt with CKD or AOCKD (*n* = 5) ▪ Vancomycin-associated kidney injury (*n* = 4) ▪ Vancomycin-associated fever (*n* = 1)
3.	As an alternative due to drug allergy	5 (29.4)	▪ Allergy to vancomycin (*n* = 5)
	**Total number**	17	

*n*: number of cases, IV: intravenous, PO: by mouth, CKD: chronic kidney disease, AOCKD: acute on chronic kidney disease. ^#^ Percentages are based on the total number of cases for each antibiotic group.

## Data Availability

The raw data supporting the conclusions of this article will be made available by the authors on request.

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
