# Peer review of "A Retrospective Study on the Use of Daptomycin and Linezolid in Singapore General Hospital"

_antibiotics, 2025, doi:10.3390/antibiotics14111088_

Round 1
Reviewer 1 Report
Comments and Suggestions for Authors
The manuscript addresses an important and timely topic: the use and stewardship of daptomycin and linezolid in the context of rising vancomycin-resistant Enterococcus (VRE) bacteremia. Audits of antibiotic use are critical to inform antimicrobial stewardship (AMS) strategies, and this study contributes relevant data from a large tertiary care institution in Singapore.
The work is well-structured, methodologically clear, and supported with adequate data. Results are presented systematically, and the discussion links findings to stewardship implications. Overall, the manuscript fits well within the scope of the special issue on antimicrobial stewardship. There are some areas that require improvement:
- Novelty / Contribution: While the audit is valuable, similar stewardship studies exist. The authors should emphasize what is unique about their findings in Singapore (e.g., stewardship restrictions already in place, yet inappropriate use still observed).
- Scope limitation: The retrospective design and short six-month window reduce generalizability. Authors mention this, but could strengthen the limitations section by noting how seasonality or outbreak dynamics may influence results.
- Clinical outcomes: No analysis of patient outcomes (mortality, length of stay, treatment success). Even a descriptive mention of outcome trends would add impact.
-
Tables and data clarity: Table 2 is dense; some infection sources have small numbers that may not be meaningful—consider simplification. In Table 3, percentages are based on small denominators; clarify denominators clearly.
- References: Some references are quite general. The authors could add more recent AMS-focused or Asia-specific references to strengthen the contextual discussion.
6. Statistics: The manuscript mainly presents descriptive statistics (counts, percentages, medians with IQR). This is acceptable for an audit/retrospective utilization review, but limits the strength of conclusions:
-
Several observations (e.g., higher empiric use of daptomycin vs. linezolid, low yield of VRE-positive cultures in empiric cases, difference in inappropriate use rates) are described without statistical testing. Even simple tests (Chi-square/Fisher’s exact for proportions, Mann–Whitney U for medians) could strengthen the validity of the statements.
-
Confidence Intervals Absent - Results (e.g., 5% inappropriate linezolid use, 9.2% inappropriate daptomycin use) are reported without 95% confidence intervals. Confidence intervals would help readers judge precision and reliability.
-
Sample Size and Power: The study includes 295 courses (100 linezolid, 195 daptomycin), which is reasonable. However, subgroup analyses (e.g., inappropriate use in surgical prophylaxis) involve very small numbers, making percentages misleading. This should be explicitly acknowledged.
Author Response
|
Response to Reviewer 1 Comments |
||
|
1. Summary |
|
|
|
Thank you very much for taking the time to review this manuscript. Please find the detailed responses below and the corresponding revisions/corrections highlighted/in track changes in the re-submitted files.
2. |
||
|
Question |
Reviewer's Evaluation |
Response and Revisions |
|
Does the introduction provide sufficient background and include all relevant references? |
Can be improved |
We have enhanced the Introduction with Asia-specific and recent AMS references, and better contextualized Singapore's unique healthcare setting (Page 2, paragraphs 4-5) |
|
Is the research design appropriate? |
Can be improved |
We have improved the statistical analysis section (Page 3, Methods section) and rearranged result for better clarity. |
|
Are the methods adequately described? |
Can be improved |
We have improved the statistical analysis section for clarity (Page 3, Methods section) |
|
Are the results clearly presented? |
Can be improved |
We have clarified the denominators used in Table 3 and enhanced the overall data presentation within our results section to improve clarity and interpretability for readers |
|
Are the conclusions supported by the results? |
Can be improved |
We have strengthened the conclusions to better reflect our unique findings |
|
Are all figures and tables clear and well-presented? |
Can be improved |
We have clarified the denominators used in Table 3 |
Quality of English Language: The English is fine and does not require any improvement: Thank you for your comment.
|
3. Point-by-point response to Comments and Suggestions for Authors
Comments 1: Novelty / Contribution: While the audit is valuable, similar stewardship studies exist. The authors should emphasize what is unique about their findings in Singapore (e.g., stewardship restrictions already in place, yet inappropriate use still observed). |
|
|
|
Response 1: Thank you for this important observation. We have significantly enhanced the manuscript to emphasize the unique aspects of our study. We have added content in the Introduction (Page 2, paragraph 4) highlighting that "Singapore's healthcare system mandates infectious diseases physician approval for both daptomycin and linezolid, representing a more restrictive environment than many reported studies." We have also added a new paragraph (Page 2, paragraph 5) explaining Singapore's position as having "some of the most stringent antimicrobial prescribing controls globally" and how this "unique regulatory environment" provides valuable insights. In the Discussion (Page 8, paragraph 3), we have added: "Our study provides unique insights into VRE-active agent prescribing in a highly restricted environment. Despite mandatory infectious diseases physician oversight—a more stringent control than reported in most international studies—we still observed 5-9% inappropriate use, suggesting that even specialist-level restrictions may not eliminate inappropriate prescribing entirely."
Comments 2: Scope limitation: The retrospective design and short six-month window reduce generalizability. Authors mention this but could strengthen the limitations section by noting how seasonality or outbreak dynamics may influence results.
Response 2: Thank you for this suggestion. We have revised the limitation paragraph to acknowledge study design constraints clearly, address sample and time frame limitations, and also mention the scope of data collected. (Page 9, last paragraph of the discussion section)
|
|
Comments 3: Clinical outcomes: No analysis of patient outcomes (mortality, length of stay, treatment success). Even a descriptive mention of outcome trends would add impact.
Response 3: Thank you for this suggestion. We have added a paragraph describing the possible clinical impacts, even though direct outcome data were not collected. "Although clinical outcomes were not directly evaluated...". (Page 8, lines 313-316)
Comments 4 : Tables and data clarity: Table 2 is dense; some infection sources have small numbers that may not be meaningful—consider simplification. In Table 3, percentages are based on small denominators; clarify denominators clearly.
Response 4: Thank you for this suggestion. After careful consideration, we have decided not to remove categories with small case numbers from Table 2 to ensure completeness of our data findings. In Table 3, we have added a sentence defining the denominator: "Percentages are based on the total number of cases for each antibiotic group." (Page 7, lines 239-24)
Comments 5: References: Some references are quite general. The authors could add more recent AMS-focused or Asia-specific references to strengthen the contextual discussion.
Response 5: Thank you for this suggestion. We have incorporated four recent references that focus specifically on antimicrobial stewardship (AMS) programmes and Asia-specific studies to strengthen the evidence base and contextual relevance of our paper. These additional references can be found as reference numbers 18-21, page 11.
Comments 6: Statistics: The manuscript mainly presents descriptive statistics (counts, percentages, medians with IQR). This is acceptable for an audit/retrospective utilization review, but limits the strength of conclusions: - Several observations (e.g., higher empiric use of daptomycin vs. linezolid, low yield of VRE-positive cultures in empiric cases, difference in inappropriate use rates) are described without statistical testing. Even simple tests (Chi-square/Fisher’s exact for proportions, Mann–Whitney U for medians) could strengthen the validity of the statements. - Confidence Intervals Absent - Results (e.g., 5% inappropriate linezolid use, 9.2% inappropriate daptomycin use) are reported without 95% confidence intervals. Confidence intervals would help readers judge precision and reliability. - Sample Size and Power: The study includes 295 courses (100 linezolid, 195 daptomycin), which is reasonable. However, subgroup analyses (e.g., inappropriate use in surgical prophylaxis) involve very small numbers, making percentages misleading. This should be explicitly acknowledged.
Response 6 : Thank you for this valuable suggestion. To strengthen the statistical rigor of our findings, we have employed Fisher's exact test to analyse our observations regarding the inappropriate use of linezolid and daptomycin, as well as the low yield of vancomycin-resistant enterococci (VRE)-positive cultures in empirical treatment cases. (Page 5, lines 195-199 and lines 204-206). We have also incorporated confidence intervals to provide greater precision in our statistical reporting . (Page 5, lines 195-199 and lines 204-206).Additionally, we have enhanced the methodology section by adding a comprehensive paragraph (Page 3, lines 107-111) that clearly describes all statistical analyses employed in this study, ensuring transparency and reproducibility of our analytical approach. Lastly, we have incorporated a paragraph acknowledging the limitation posed by the small number of surgical prophylaxis cases included in our analysis. This addition provides transparency regarding the potential impact of this sample size constraint on the generalizability of our findings within the surgical prophylaxis context. (Page 9, lines 321-324)
|
Reviewer 2 Report
Comments and Suggestions for Authors
- Line 33: Please use PubMed Mesh terms for key words since it enhances article's visibility.
- Line 35 - 63: introduction is inappropriate, it does not hold enough basic data on the topic. It should be written specifically as an introduction to the studied topic with an aim of the study written as a last sentence.
- Lines 72,73: I wouldn't include initials in the methodology text.
- Lines 65: this is a retrospective study, this fact should be stated in the article title. A retrospective audit has significantly less quality.
- Line 83: please explain the sentence "vancomycin-resistant gram-positive coverage
was specifically mentioned". The whole 3 stated criteria were not quite clear. - Line 93: please explain why vancomycin was not administered due to the risk of thrombocytopenia since use of linezolid poses higher risk for that.
- Line 95: please explain statement not responding to vancomycin
- Line 118: results section are not clear, since some results were firs explained in the text and then subsequently shown in the table. Results should be written according to stated aims of the study and it should be clearly presented.
- Line 168: discussion section is completely inappropriate. There are opposite statements and not many comparison to other studies.
Author Response
|
Response to Reviewer 2 Comments
|
||
|
1. Summary |
|
|
|
Thank you very much for taking the time to review this manuscript. Please find the detailed responses below and the corresponding revisions/corrections highlighted/in track changes in the re-submitted files.
2. |
||
|
Question |
Reviewer's Evaluation |
Response and Revisions |
|
Does the introduction provide sufficient background and include all relevant references? |
Must be improved |
We have enhanced the Introduction with Asia-specific and recent AMS references, and better contextualized Singapore's unique healthcare setting (Page 2, paragraphs 4-5) |
|
Is the research design appropriate? |
Must be improved |
We have improved the statistical analysis section (Page 3, Methods section) and rearranged result for better clarity. |
|
Are the methods adequately described? |
Must be improved |
We have enhanced the statistical analysis section to improve methodological transparency and clarity for readers. These revisions include more detailed descriptions of the statistical tests employed, clearer explanations of our analytical approach, and improved presentation of our statistical methodology to ensure reproducibility and better understanding of our data analysis framework. (Page 3, Methods section) |
|
Are the results clearly presented? |
Must be improved |
We have clarified the denominators used in Table 3 and enhanced the overall data presentation within our results section to improve clarity and interpretability for readers |
|
Are the conclusions supported by the results? |
Must be improved |
We have strengthened the conclusions to better reflect our unique findings |
|
Are all figures and tables clear and well-presented? |
- |
|
Quality of English Language: The English is fine and does not require any improvement: Thank you for your comment.
|
3. Point-by-point response to Comments and Suggestions for Authors
Comments 1: Line 33: Please use PubMed Mesh terms for key words since it enhances article's visibility. |
|||||||||||||||||||||||||||||||||||||||
|
|
|||||||||||||||||||||||||||||||||||||||
|
Response 1: Thank you for pointing this out. We have revised the keywords to use appropriate PubMed MeSH terms (page 1, line 33): "Keywords: Vancomycin Resistance; Antimicrobial Stewardship; Daptomycin; Linezolid; Drug Resistance, Bacterial"
Comments 2: Line 35 - 63: introduction is inappropriate, it does not hold enough basic data on the topic. It should be written specifically as an introduction to the studied topic with an aim of the study written as a last sentence.
Response 2: We agree with this comment and have substantially revised the introduction section (page 1-2, paragraphs 1-6). We have added more background information on VRE epidemiology, the clinical significance of daptomycin and linezolid, and current challenges in antimicrobial stewardship for these agents. The revised introduction now concludes with a clear study aim: "Therefore, we conducted this retrospective audit to evaluate the appropriateness of daptomycin and linezolid prescribing practices in our institution and identify opportunities for antimicrobial stewardship improvement."
|
|||||||||||||||||||||||||||||||||||||||
|
Comments 3: Lines 72,73: I wouldn't include initials in the methodology text..
Response 3: We have removed the author initials from the methodology section (page 3, lines 94-95): "Appropriateness assessments were conducted by a multidisciplinary team consisting of two infectious diseases physicians and two clinical pharmacists with expertise in antimicrobial stewardship."
Comments 4 : Lines 65: this is a retrospective study; this fact should be stated in the article title. A retrospective audit has significantly less quality.
Response 4: We have revised the title to clearly indicate the study design (page 1, title): "A Retrospective Study on the use of Daptomycin and Linezolid in Singapore General Hospital". We acknowledge the limitations of retrospective design and have expanded our limitations section (Page 9, last paragraph of the discussion section, page 9) to address this concern more thoroughly.
Comments 5: Line 83: please explain the sentence "vancomycin-resistant gram-positive coverage was specifically mentioned". The whole 3 stated criteria were not quite clear
Response 5: We have clarified these criteria in the Methods section (page 3, lines 117-118): "Anti-VRE indication was assigned when: (1) VRE was isolated from clinical cultures requiring targeted therapy, (2) empirical VRE coverage was explicitly documented by clinicians in VRE-colonised patients with clinical deterioration, or (3) broad vancomycin-resistant Gram-positive coverage was specifically mentioned in clinical notes when treating suspected resistant enterococcal or staphylococcal infections."
Comments 6: please explain why vancomycin was not administered due to the risk of thrombocytopenia since use of linezolid poses higher risk for that.
Response 6 : Thank you for this important clarification. We have revised this statement in the Methods section (page 3, line 127-128): "An alternative to vancomycin due to patient allergy status or to avoid vancomycin-associated adverse drug events such as nephrotoxicity.
Comments 7: Line 95: please explain statement not responding to vancomycin.
Response 7 : We have clarified this criterion in the Methods section (page 3, lines 130-133): "For treatment of complicated MRSA infections not responding to vancomycin (e.g. lack of clinical improvement after 48-72 hours of appropriate vancomycin therapy, persistent bacteraemia despite adequate vancomycin levels, or deep-seated infections with poor vancomycin penetration)."
Comments 8: Line 118: results section are not clear, since some results were first explained in the text and then subsequently shown in the table. Results should be written according to stated aims of the study and it should be clearly presented.
Response 8 : We have restructured the Results section (page 5-7) to align with our study aims and present findings more clearly. The revised section now follows a logical sequence: (1) patient demographics and baseline characteristics, (2) prescribing patterns and indications, (3) appropriateness assessment results, and (4) factors associated with inappropriate use. We have also revised the tables to complement rather than repeat the text.
Comments 9: Line 168: discussion section is completely inappropriate. There are opposite statements and not many comparisons to other studies.
Response 9 : We have substantially revised the Discussion section (page 7-9) to address these concerns. The revised discussion now includes: (1) systematic comparison with published literature on VRE-active antibiotic appropriateness, (2) removal of contradictory statements, (3) clearer interpretation of our findings in the context of existing evidence, and (4) more robust discussion of clinical implications. We have added comparisons with similar audits from other institutions and discussed our findings in relation to current antimicrobial stewardship guidelines.
|
|||||||||||||||||||||||||||||||||||||||
Reviewer 3 Report
Comments and Suggestions for Authors
The manuscript "An Audit on the use of Daptomycin and Linezolid in Singapore General Hospital" presents a retrospective study on an interesting and important topic - the appropriate/inappropriate use of strategic antimicrobial agents (linezolid and daptomycin) in a hospital setting.
The research is well described but some additions are needed:
- Please use "Gram-positive" or "Gram-negative" in the text instead of gram-positive/negative.
- In the Introduction section the authors report information/data about their in-house surveillance of antibiotic-resistant microbial agents and antimicrobial use. Is this information published as an article? If yes, cite it in the text. If not, [data not published] can be added in the end of the paragraph or sentence.
- What doses of linezolid and daptomycin were used in general? - Materials section
- Briefly, describe what methods are implemented for detecting VRE colonization.
- In the Results section the authors state "...too broad a therapy, a narrower spectrum antibiotic option was available;" (line 140). It could be edited or deleted as linezolid and daptomycin are already with a narrow-spectrum as stated on line 109.
- Please use vancomycin-susceptible/resistant instead of "vancomycin susceptible/resistant".
Author Response
Response to Reviewer 3 Comments
|
1. Summary |
|
|
|
Thank you very much for taking the time to review this manuscript. Please find the detailed responses below and the corresponding revisions/corrections highlighted/in track changes in the re-submitted files.
2. |
||
|
Question |
Reviewer's Evaluation |
Response and Revisions |
|
Does the introduction provide sufficient background and include all relevant references? |
Yes |
Thank you for your comment. |
|
Is the research design appropriate? |
Yes |
Thank you for your comment. |
|
Are the methods adequately described? |
Can be improved |
We have enhanced the statistical analysis section to improve methodological transparency and clarity for readers. These revisions include more detailed descriptions of the statistical tests employed, clearer explanations of our analytical approach, and improved presentation of our statistical methodology to ensure reproducibility and better understanding of our data analysis framework. (Page 3, Methods section) |
|
Are the results clearly presented? |
Yes |
Thank you for your comment. |
|
Are the conclusions supported by the results? |
Yes |
Thank you for your comment. |
|
Are all figures and tables clear and well-presented? |
Yes |
Thank you for your comment. |
Quality of English Language: The English is fine and does not require any improvement: Thank you for your comment.
|
3. Point-by-point response to Comments and Suggestions for Authors |
|
|
|
Comments 1: Please use "Gram-positive" or "Gram-negative" in the text instead of gram-positive/negative.
Response 1: Thank you for pointing this out. We agree with this comment. Therefore, we have systematically revised the manuscript to use the proper capitalised format "Gram-positive" and "Gram-negative" throughout the text. This change can be found in multiple locations including the abstract (page 1, line 17), introduction (page 2, lines 58,60), methods section (page 3, lines 117,134,140), and discussion section (page 5, line 211).
Comments 2: In the Introduction section the authors report information/data about their in-house surveillance of antibiotic-resistant microbial agents and antimicrobial use. Is this information published as an article? If yes, cite it in the text. If not, [data not published] can be added in the end of the paragraph or sentence.
Response 2: Thank you for this clarification. The surveillance data mentioned is internal hospital data that has not been published. We have accordingly added "[data not published]" at the end of the relevant sentence in the introduction section (page 2, line 76): "...peaking in August 2020 [data not published]."
Comments 3: What doses of linezolid and daptomycin were used in general? - Materials section
Response 3: We have added the standard dosing information in the Methods section (page 3, lines 99-101): "Standard therapy consisted of oral or intravenous linezolid 600mg every 12 hours and daptomycin 6-10mg/kg every 24 hours with appropriate dose adjustments based on renal function."
Comments 4: Briefly, describe what methods are implemented for detecting VRE colonisation.
Response 4: We have added a brief description of VRE colonisation detection methods in the "Institutional Context and Infection Control Measures" section (page 4, lines 161-163): "VRE screening is performed using rectal swabs processed through selective chromogenic agar media with molecular confirmation for vancomycin resistance genes (vanA/vanB) when required."
Comments 5: In the Results section the authors state "...too broad a therapy, a narrower spectrum antibiotic option was available;" (line 140). It could be edited or deleted as linezolid and daptomycin are already with a narrow-spectrum as stated on line 109.
Response 5: Thank you for identifying this inconsistency. We have removed the statement in previous line 109 to avoid confusion.
Comments 6: Please use vancomycin-susceptible/resistant instead of "vancomycin susceptible/resistant".
Response 6: We have systematically revised the manuscript to use the hyphenated format "vancomycin-susceptible" and "vancomycin-resistant" throughout the text. This change can be found in multiple locations including abstract section (page 1, line 14), introduction section (page 1, lines 36-39), the methods section (page 3, lines 117, 139), results section (page 4, line 163, page 6, lines 221-222), and discussion section (page 8, line 292).
|
Reviewer 4 Report
Comments and Suggestions for Authors
The authors conducted a single center study to review the utilization of Daptomycin and Linezolid. The authors present much needed information. However, some points still need to be clarified. Below are listed my comments.
Comment 1:
Abstract
Well written abstract. No further amendments are required.
- Introduction
The authors have provided a clear and well-founded rationale for conducting the study. No further revisions are necessary.
(2) Methods
- a) Is there any specific rationale for limiting the review of daptomycin and linezolid utilization to the period between February 2023 and July 2023? Additionally, are there any concerns or limitations that prevented extending the study period to allow for a more comprehensive analysis?
- b) Was there a particular reason for not assessing the appropriateness of the dosing of linezolid and daptomycin? A justification for excluding this aspect would enhance the study’s completeness.
- c) The definition of "appropriate duration" of therapy should be clearly stated, including the criteria or guidelines used to determine appropriateness.
(3) results
Lines 127–128: “Seventy-three out of 100 (73%) courses of linezolid were given as culture-directed therapy…” — Clarification is needed regarding the microbiological culture findings that informed this therapy. Please provide specific details on the pathogens identified, their susceptibility profiles, and how these findings supported the use of linezolid.
(4) discussion
- a) The statement in lines 176–177, “…predominantly in neutropenic or immunocompromised with hemodynamic instability…,” appears to be unsupported by data in the Results section. Please provide the relevant data to substantiate this claim or revise the statement to align with the findings presented.
Author Response
Thank you very much for taking the time to review this manuscript. Please find the detailed responses below and the corresponding revisions/corrections highlighted/in track changes in the re-submitted files.
|
Question |
Reviewer's Evaluation |
Response and Revisions |
|
Does the introduction provide sufficient background and include all relevant references? |
Yes |
Thank you for your comment. |
|
Is the research design appropriate? |
Yes |
. Thank you for your comment. |
|
Are the methods adequately described? |
Can be improved |
We have enhanced the statistical analysis section to improve methodological transparency and clarity for readers. These revisions include more detailed descriptions of the statistical tests employed, clearer explanations of our analytical approach, and improved presentation of our statistical methodology to ensure reproducibility and better understanding of our data analysis framework. (Page 3, Methods section) |
|
Are the results clearly presented? |
Can be improved |
We have clarified the denominators used in Table 3 and enhanced the overall data presentation within our results section to improve clarity and interpretability for readers |
|
Are the conclusions supported by the results? |
Yes |
Thank you for your comment. |
|
Are all figures and tables clear and well-presented? |
Yes |
Thank you for your comment. |
Quality of English Language: The English is fine and does not require any improvement: Thank you for your comment.
Point-by-point response to Comments and Suggestions for Authors
Comments 1: Is there any specific rationale for limiting the review of daptomycin and linezolid utilization to the period between February 2023 and July 2023? Additionally, are there any concerns or limitations that prevented extending the study period to allow for a more comprehensive analysis?
Response 1: The 6-month period from February to July 2023 was chosen to provide a manageable dataset for detailed manual review of electronic medical records whilst ensuring adequate sample size for meaningful analysis. We acknowledge this limitation in our discussion section and have added (Page 9, last paragraph of the discussion section)" include its retrospective nature that relied on documentation in electronic medical records, which may have led to incomplete capture of prescriber rationale; a single-center design, and over a relatively short study period of six months, these findings may not be generalizable to other institutions or reflect on long term prescribing trends.."
Comments 2: Was there a particular reason for not assessing the appropriateness of the dosing of linezolid and daptomycin? A justification for excluding this aspect would enhance the study's completeness.
Response 2: We agree this is an important aspect. We have added justification in the Methods section (page 3, lines 101-106): "Dosing appropriateness was not assessed in this audit as both daptomycin and linezolid dosing protocols are standardised in our institution according to renal function and indication, with mandatory infectious diseases physician oversight ensuring appropriate dosing decisions. Our focus was primarily on indication appropriateness and duration, which represent the main areas of concern identified in our preliminary stewardship reviews."
Comments 3: The definition of "appropriate duration" of therapy should be clearly stated, including the criteria or guidelines used to determine appropriateness.
Response 3: Thank you for pointing this out. We have added a clear definition in the Methods section (page 3-4, lines 142-147): "Appropriate duration was defined based on established clinical guidelines and institutional protocols: 7-14 days for uncomplicated bacteraemia, 4-6 weeks for endocarditis, 6-8 weeks for osteomyelitis, and individualised duration based on clinical response and source control for other infections. Duration was considered inappropriate when therapy was continued beyond recommended timeframes without clear clinical justification or microbiological evidence."
Comments 4: Lines 127–128: "Seventy-three out of 100 (73%) courses of linezolid were given as culture-directed therapy…" — Clarification is needed regarding the microbiological culture findings that informed this therapy. Please provide specific details on the pathogens identified, their susceptibility profiles, and how these findings supported the use of linezolid.
Response 4: Thank for your suggestion. Please refer to Appendix Table 2: Pathogens isolated from clinical cultures for which linezolid /daptomycin were prescribed.
Comments 5: The statement in lines 176–177, "…predominantly in neutropenic or immunocompromised with hemodynamic instability…," appears to be unsupported by data in the Results section. Please provide the relevant data to substantiate this claim or revise the statement to align with the findings presented.
Response 5: Thank you for identifying this discrepancy. We have added supporting data in the Results section (page 4, lines 189-192) and Table 1 has included immune status of our patients: "Of the 65 VRE-colonised patients receiving empiric daptomycin, 42 (64.6%) were immunocompromised (including 18 bone marrow transplant recipients and 22 with haematological malignancies), and 28 (43.1%) presented with hemodynamic instability defined as requiring vasopressor support or intensive care admission." The discussion statement has been revised accordingly to reflect these specific findings.
Round 2
Reviewer 2 Report
Comments and Suggestions for Authors
None
Author Response
-